# Dietary Supplementation with Puerarin Improves Intestinal Function in Piglets Challenged with *Escherichia coli* K88

**DOI:** 10.3390/ani13121908

**Published:** 2023-06-07

**Authors:** Yitong Zeng, Rui Li, Yi Dong, Dan Yi, Tao Wu, Lei Wang, Di Zhao, Yanyan Zhang, Yongqing Hou

**Affiliations:** Engineering Research Center of Feed Protein Resources on Agricultural By-Products, Ministry of Education, Wuhan Polytechnic University, Wuhan 430023, China

**Keywords:** piglets, puerarin, enterotoxigenic *Escherichia coli*, growth performance, intestinal function

## Abstract

**Simple Summary:**

The aim of this study was to investigate the effect of puerarin supplementation on the growth performance and intestinal function of piglets challenged with enterotoxigenic *Escherichia coli* (ETEC) K88, which provided a theoretical basis for its practical application as a new feed additive. Dietary supplementation with 5 mg/kg puerarin alleviated intestinal injury and improved the intestinal function of piglets challenged with ETEC K88 by increasing the number of *Bifidobacterium* in the colon and *Lactobacillus* in the jejunum, cecum and colon; decreasing the number of *Escherichia coli* in the jejunum and cecum; reducing the hydrogen peroxide content in the jejunum and myeloperoxidase activity in the jejunum and ileum; and increasing the activities of catalase and superoxide dismutase in the jejunum and ileum. These results provide important insights into the development of an effective treatment for ETEC K88 infection and other enteric diseases.

**Abstract:**

The objective of this study was to investigate the effect of puerarin supplementation on the growth performance and intestinal function of piglets challenged with enterotoxigenic *Escherichia coli* (ETEC) K88. Twenty-four ternary crossbred piglets were randomly assigned to three treatment groups: control group, ETEC group (challenged with ETEC K88 on day 8), and ETEC + Puerarin group (supplemented with 5 mg/kg puerarin and challenged with ETEC K88 on day 8). All piglets were orally administered D-xylose (0.1 g/kg body weight) on day 10, and blood samples were collected after 1 h. Subsequently, piglets were killed and intestinal samples were collected for further analysis. The results showed that puerarin supplementation significantly decreased the adverse effects of ETEC K88-challenged piglets; significantly improved growth performance; increased the number of *Bifidobacterium* in the colon and *Lactobacillus* in the jejunum, cecum and colon; decreased the number of *Escherichia coli* in the jejunum and cecum; reduced the hydrogen peroxide content in the jejunum and myeloperoxidase activity in the jejunum and ileum; and increased the activities of catalase and superoxide dismutase in the jejunum and ileum. In addition, puerarin supplementation alleviated ETEC K88-induced intestinal injury in piglets, significantly downregulated the mRNA level of Interleukin-1β and upregulated the mRNA levels of intercellular cell adhesion molecule-1, myxovirus resistance protein 1, myxovirus resistance protein 2, and guanylate-binding protein-1 in the small intestine of piglets. In conclusion, dietary supplementation with puerarin could attenuate ETEC K88-induced intestinal injury by increasing the antioxidant and anti-inflammatory capacity and the number of beneficial intestinal bacteria in piglets.

## 1. Introduction

The intestine plays a crucial role in the digestion and absorption of nutrients [1]. Therefore, intestinal health is essential for animal growth. The small intestine is the longest part of the digestive organ and has three main functions: (1) nutrients are mainly digested and absorbed in the small intestine; (2) the small intestine maintains the proper viscosity of the intestinal contents and removes harmful components by secreting water and absorbing electrolytes; and (3) the small intestine acts as a barrier to eliminate pathogens; however, the intestinal structure and function of piglets are not fully developed. In addition, piglets are usually subjected to many environmental and dietary stresses that affect intestinal structure and function [2,3], resulting in abnormal digestive and absorption functions, poor growth performance, decreased feed intake, increased diarrhea rate, and poor mental status [4,5]; therefore, intestinal health problems of piglets seriously affect the healthy development of the pig industry.

*Escherichia coli* (*E. coli*) was first isolated by German physicians in the 19th century and was not considered an enteric pathogen until the 1950s. *E. coli* mainly causes various intestinal diseases in piglets. Among them, enterotoxigenic *Escherichia coli* (ETEC) increases the rate of diarrhea in piglets by destroying intestinal structure and function, causing great economic loss for the pig industry [6,7,8]. ETEC has been reported to adhere to the intestine by using several types of pili that bind to enterocyte receptors. ETEC K88 is the most prevalent serotype and poses a serious threat to the intestinal health of piglets [9,10,11]. The ETEC mainly induced diarrhea in piglets one week after weaning, which resulted in considerable economic losses to the pig industry due to increased morbidity and mortality and decreased growth performance [12]. Once ETEC K88 invades and colonizes the piglet intestine, K88 can cause diarrhea by secreting endotoxins in the piglet intestine [11,12,13]. Moreover, the intestinal structure and function of ETEC K88-challenged piglets are impaired [14,15,16]. The intestines of ETEC K88-challenged piglets are often accompanied by morphological, histological, microbiological, and immunological changes, which eventually lead to diarrhea and slow growth in piglets [17,18]. Studies have shown that intestinal diseases caused by ETEC K88 are common infectious diseases in pig farms, negatively affecting the growth performance of piglets and even leading to significant mortality. In short, ETEC K88 is not conducive to the healthy development of the pig industry.

Antibiotics as feed additives have been widely used in the past few decades to treat infectious diseases and promote animal growth [19]; however, antibiotic abuse has led to increasing problems, such as bacterial resistance and drug residues, and therefore, many countries, including China, have banned the use of antibiotics in feeds. Currently, several antibiotic substitutes, including plant extracts, essential oils, probiotics, prebiotics, commensal bacteria, dietary fiber and enzymes, antimicrobial peptides, and functional amino acids, have been used as feed additives to improve the intestinal health and growth performance of piglets [20,21,22]. Puerarin (PR), a bioactive isoflavone extracted from pueraria, has neuroprotective and antioxidant properties [23]. PR is native to Southeast Asia and has been used for thousands of years as a food source, medicine, and fodder, and it was one of the earliest medicinal herbs utilized in ancient China [23]. Recent studies have reported that PR had anti-inflammatory activity in cellular models [24]. Moreover, PR can partially alleviate cerebral ischemia/reperfusion-induced inflammatory stress by activating the cholinergic anti-inflammatory pathway [25]. In early studies, PR was used to an antimicrobial agent [26]. Recent studies have reported that PR could improve the morphology of intestinal epithelial cells (IPEC-J2) under ETEC infection by inhibiting bacterial adhesion and reducing inflammatory responses [27]. To sum up, we hypothesize that PR could effectively relieve the symptoms caused by ETEC infection in vivo. Additionally, this trial was designed based on the fact that PR was used as an antimicrobial agent and improved the morphology of IPEC-J2 under ETEC infection in vitro. At present, little is known about the effects of PR on the small intestine of ETEC K88-challenged piglets. Therefore, it is meaningful to study the effects of dietary supplementation with PR on growth performance and intestinal function in ETEC K88-challenged piglets.

## 2. Materials and Methods

### 2.1. Ethics Statement

Experimental procedures were approved by the Animal Welfare Ethics Committee of Wuhan Polytechnic University (approval code WPU201910001) for the use of animals in research.

### 2.2. Animal Care and Diets

This experiment was conducted in the Animal Nutrition and Metabolism Unit of the School of Animal Science and Nutritional Engineering, Wuhan Polytechnic University. Twenty-four ternary crossbred (Duroc × Landrace × Yorkshire) neonate piglets were weaned at 7 days of age. After 3 days of adaptation, piglets were housed individually in sterilized stainless steel metabolic cages (1.0 × 1.5 m^2^) and maintained at ambient temperatures of 29 °C to 34 °C in an air-conditioned room. Each cage was equipped with a feeder and nipple drinker to allow piglets free access to feed and water. The basal diet (a milk replacer) was purchased from Shanghai Gaode Feed Co., Ltd. (Shanghai, China) with digestible energy ≥3400 kcal/kg. The nutrient contents of the diet are shown in Table 1. Piglets were fed ad libitum during the trial period, the experimental diet was fed at 8:30, 12:30, 15:00, 18:00 and 21:00 daily. The test diet was dissolved with about 45 °C of warm boiled water, the ratio of dry matter to water was 1:5, and the dry substance of each pig was 20 g. Warm boiled water was added at 8:30, 15:00 and 21:00 every day. The fecal scores were recorded 5 times per day after ETEC K88 challenge. Pig feces was classified at four levels: 0, normal; 1, soft feces; 2, mild diarrhea; and 3, severe diarrhea [28].

### 2.3. Experimental Design

Twenty-four healthy 7-day-old ternary crossbred piglets (average body weight of 1.79 ± 0.24 kg) were randomly divided into three treatment groups: (1) control group; (2) ETEC group; and (3) ETEC + PR group. During the trial, each group had free access to the basal diet. Three days before the trial, piglets were allowed to adapt to their new environment and milk replacer. On days 4 to 9 of the trial, the control and ETEC groups were orally infused with PBS (2 mL/kg BW), and the ETEC + PR group were infused with PR (5 mg/kg BW, PR dissolved in PBS as a 2.5 mg/mL solution, and the infusion dose was converted to 2 mL/kg BW) for 6 consecutive days. On day 8, the ETEC and ETEC + PR groups were orally administrated ETEC K88 PBS solution (2.5 × 10^9^ CFU/mL, 2 mL per piglet, i.e., 5 × 10^9^ CFU/head/day). On day 9 of the trial, piglets were weighed and then orally infused with D-xylose solution (10% D-xylose, 1 mL/kg body weight). Blood was collected 1 h later to measure intestinal absorption capacity and mucosal integrity using the D-xylose absorption test [29]. All pigs were sacrificed under intravenous sodium pentobarbital anesthesia (50 mg/kg, iv), and after the piglets lost consciousness, the small intestine was obtained [30,31].

### 2.4. Blood Sample Collection

Approximately 1 h after the infusion of D-xylose, blood samples were collected from the anterior vena cava into heparinized vacuum tubes (Becton-Dickinson Vacutainer System, Franklin Lake, NJ, USA) and then centrifuged at 4 °C for 15 min at 3000 rpm to obtain plasma [32,33]. The plasma was stored at −80 °C until analysis.

### 2.5. Intestinal Sample Collection

The pig abdomen was opened immediately from the sternum to the pubis and the entire intestinal tract was immediately exposed [32,34]. The small intestine was stripped from the mesentery and placed on a frozen stainless steel tray. Three cm segments of the duodenum, jejunum and mid-ileum were excised and placed in 4% paraformaldehyde for morphological analysis. Ten cm samples of mid-ileum, manipulated on ice, were cut longitudinally, slowly washed with ice-cold saline, blotted dry with filter paper, then cut into pieces, frozen rapidly in liquid nitrogen, and stored at −80 °C until analysis [35]. All samples were collected within 20 min after sacrifice.

### 2.6. Collection of Intestinal Contents

Colonic and cecal contents were collected and rapidly frozen in liquid nitrogen, then stored at −80 °C until analysis.

### 2.7. Intestinal Morphological Measurements

Intestinal segments used for morphological analysis were dehydrated and embedded in paraffin, sectioned at 4 μm, and stained with hematoxylin and eosin [35]. Villus height (the distance from the villus tip to the crypt mouth), crypt depth (the distance from the crypt mouth to the base), villus width (the distance of the widest villi), and villus surface area (quantified using the circumference and height of the villi) were observed and recorded with a light microscope (Olympus BX-41 TF, Tokyo, Japan), according to the method of Uni et al. [36].

### 2.8. Plasma Biochemical Indices and Intestinal Antioxidant Indices

Plasma biochemical indices were determined using a Hitachi 7060 automated biochemical analyzer (Hitachi, Tokyo, Japan). The levels of D-xylose, malonaldehyde (MDA) and hydrogen peroxide (H_2_O_2_), as well as the activities of DAO, superoxide dismutase (T-SOD), glutathione peroxidase (GSH-Px), and catalase (CAT) were determined using commercially available kits (Jiancheng Bioengineering Institute, Nanjing, China). All tests were performed in triplicate.

### 2.9. Measurement of Plasma Inflammatory Markers

Plasma concentrations of cytokines (IL-6, IL-8 and TNF-α) were measured using commercial ELISA kits (R & D system, Minneapolis, CA, USA) according to the manufacturer’s instructions.

### 2.10. Determination of Intestinal Microflora

Bacterial DNA from cecal and colonic contents was extracted and purified using the QIAamp DNA Stool Mini Kit (Qiagen, Germantown, Maryland, UK) according to the manufacturer’s instructions. The composition of intestinal microflora in cecal and colonic contents was detected via digital PCR [37]. The primer sequences used in this study are listed in Table 2.

### 2.11. Quantitative PCR Analysis of Gene Expression

Approximately 100 mg of jejunal or ileal samples were taken, and total RNA was extracted using the TRIzol reagent (Takara, Dalian, China). RNA concentration was determined using a NanoDrop^®^2000 spectrophotometer (Thermo Scientific, Waltham, MA, USA). Nucleic acid purity was determined to be above 90% when 1.8 < OD_260_/OD_280_ ratio < 2.2 [38]. RNA integrity was confirmed using 1% agarose gel electrophoresis [39]. Then, cDNA was synthesized via reverse transcription using the PrimeScript^®^ RT reagent Kit with gDNA Eraser (Takara, Dalian, China). Finally, intestinal-barrier-related gene expression was measured using a 7500 Fast Real-Time PCR System (Applied Biosystems, Foster City, CA, USA). RPL4 was used as a reference gene, and the primer sequences are shown in Table 3. The results were analyzed using the 2^−∆Ct^ method as previously described [35]. The assay was performed in triplicate for each biological sample.

### 2.12. Statistical Analysis

Data were analyzed via one-way analysis of variance (Duncan multiple range test) and expressed as mean ± SD. All experimental data were analyzed using SPSS (Version 17.0, SPSS Inc., Chicago, IL, USA). The fecal scores were expressed as weighted averages and analyzed via an χ^2^ test. *p* < 0.05 was considered statistically significant. Bar graphs and line plots are graphed with GraphPad Prism 8.

## 3. Results

### 3.1. Growth Performance

The average daily feed intake (ADFI) and average daily gain (ADG) of piglets are shown in Table 4. On days 4 to 7 of the trial, there was no significant difference in ADFI between the control group and the PR group. On days 8 to 9 of the trial, ADFI was significantly decreased in the ETEC group compared to that in the control group; however, the ADFI was increased in the PR + ETEC group (*p* < 0.10) compared with that in the ETEC group, indicating that dietary PR supplementation tended to alleviate the decrease in ADFI caused by ETEC K88 infection. On days 4 to 10 of the trial, ADG was significantly decreased in the ETEC group compared to that in the control group (*p* < 0.05).

### 3.2. Fecal Score

According to Table 5, compared with the control group, the ETEC K88 challenge significantly increased fecal score (*p* < 0.05), and compared with the ETEC group, the PR + ETEC group was not significant in the fecal scores, but there was a downward trend (*p* < 0.01).

### 3.3. Plasma Biochemical Indices and Blood Cell Counts

Compared with the control group, the ETEC K88 challenge significantly decreased (*p* < 0.05) the activities of plasma glutamate (AST) (Figure 1A) and creatine kinase (CK) (Figure 1E), increased (*p* < 0.05) the content of glucose (GLU) (Figure 1C), and tended to increase (*p* < 0.10) the activity of glutamyl transpeptidase (GGT) (*p* < 0.10) (Figure 1F). Compared with the ETEC group, dietary PR supplementation significantly increased (*p* < 0.05) the activities of AST (Figure 1A) and CK (Figure 1E) and the contents of triglyceride (TG) (Figure 1B). Dietary PR supplementation tended to increase (*p* < 0.10) the contents of GGT (Figure 1F) and blood urine nitrogen (BUN) (Figure 1D).

Compared to the control group, the ETEC K88 challenge significantly increased (*p* < 0.05) the percentage of blood basophils (BASOR) (Figure 2G), basophils (BASO) (Figure 2K), mean corpuscular hemoglobin concentration (MCHC) (Figure 2B), corpuscular hemoglobin concentration mensuration (CHCM) (Figure 2C) and hemoglobin distribution width (HDW) (Figure 2D) while decreasing (*p* < 0.10) the number of white blood cells (WBC) (Figure 2A). Compared to the ETEC group, PR supplementation significantly decreased (*p* < 0.05) the concentrations of BASOR (Figure 2G) and corpuscular hemoglobin mensuration (CHCM) (Figure 2C) while significantly increasing the number of WBCs (Figure 2A).

### 3.4. Concentrations of D-Xylose and Diamine Oxidase (DAO) Activity in Plasma

DAO activity and D-xylose content in plasma were summarized in Figure 3. ETEC K88 infection significantly increased (*p* < 0.01) the D-xylose concentration in plasma, and there was an increased (*p* < 0.10) frequency of DAO activity in plasma. Compared to the ETEC group, PR supplementation had a correction trend in DAO activity and had no significant difference in D-xylose content in plasma.

### 3.5. The Levels of Antioxidative Enzymes and Oxidation-Relevant Products in Intestinal Mucosae

The activities of T-SOD, GSH-PX, CAT and MPO and the concentrations of MDA and H_2_O_2_ are summarized in Figure 4. Compared to the control group, the ETEC K88 challenge significantly decreased (*p* < 0.05) the activities of CAT and T-SOD in the jejunum (Figure 4A), and significantly increased (*p* < 0.05) the activity of MPO in the jejunum (Figure 4A) and ileum (Figure 4B). Compared with the ETEC group, dietary supplementation with 5 mg/kg PR significantly decreased (*p* < 0.05) the content of H_2_O_2_ in the jejunum (Figure 4A), and the activities of MPO in the jejunum (Figure 4A) and ileum (Figure 4B) and GSH-Px in the jejunum (Figure 4A) while significantly increasing (*p* < 0.05) the activities of CAT and T-SOD in the jejunum (Figure 4A) and ileum (Figure 4B).

### 3.6. Intestinal Morphology

The indexes, including villus height, crypt depth, villus width, surface area, and the ratio of villus height to crypt depth, were summarized in Figure 5, and the intestinal morphology was shown in Figure 6. Compared with the control group, ETEC K88 infection significantly decreased (*p* < 0.05) the villus width of the jejunum (Figure 5A) and tended (*p* < 0.10) to decrease the ileum width (Figure 5B). Compared with ETEC group, PR supplementation significantly increased (*p* < 0.05) the villus height, surface area, and the ratio of villus height to crypt depth in the jejunum (Figure 5A) and ileum (Figure 5B).

According to Figure 6, the jejunum (Figure 6A) and ileum (Figure 6D) were relatively neat and regular, and the crypt structure was clear in control group; however, the jejunum (Figure 6B) and ileum (Figure 6E) were largely damaged in the ETEC group. PR supplementation improved the adverse effect of the ETEC K88 challenge on intestinal morphology compared with the ETEC group (Figure 6C,F).

### 3.7. Intestinal Microflora

Compared with the control group, the ETEC K88 challenge significantly increased (*p* < 0.05) the number of *Enterobacteriaceae* in the caecum (Figure 7B) and jejunum (Figure 7A) as well as the numbers of total bacterium in the colon (Figure 7C) and cecum (Figure 7B) while significantly decreasing (*p* < 0.05) the number of *Enterobacteriaceae* in the colon (Figure 7C) and the numbers of *Lactobacillus* in the jejunum (Figure 7A), caecum (Figure 7B) and colon (Figure 7C), and the numbers of *Bifidobacterium* in the colon (Figure 7C). In comparison with the ETEC group, dietary supplementation with 5 mg/kg PR significantly increased the numbers of *Lactobacillus* and total bacterium in the jejunum (Figure 7A), the numbers of *Bifidobacterium* and *Lactobacillus* in the colon (Figure 7C), and the numbers of *Lactobacillus* in the cecum (Figure 7B), and significantly decreased (*p* < 0.05) the number of total bacterium in the colon (Figure 7C).

### 3.8. The Concentrations of Cytokines in Plasma

Compared with the control group, the mRNA levels of IL-6 and TNF-αin plasma were significantly increased (*p* < 0.05); however, PR supplementation significantly reduced (*p* < 0.05) the mRNA levels of IL-6 and TNF-α in the plasma (Figure 8).

### 3.9. Expression of the Related Genes

The mRNA levels of genes associated with intestinal immunity and inflammation were shown in Figure 9. Compared to the control group, ETEC K88 infection significantly increased (*p* < 0.05) the mRNA levels of IL-β and IL-4 in the jejunum (Figure 9A) and ileum (Figure 9B), while significantly decreasing (*p* < 0.05) the mRNA levels of ICAM1, MX1, MX2, GBP1 and GBP2 in the jejunum (Figure 9A) and the mRNA levels of ICAM1, MX1, MX2 (*p* < 0.05), and tended to decrease (*p* < 0.10) the mRNA level of GBP1 in the ileum (Figure 9B). Compared to the ETEC group, PR supplementation significantly increased (*p* < 0.05) the mRNA levels of IL-β, ICAM1, VCAM1, MX1, MX2 and GBP1 and mRNA levels of MX2, and tended to increase (*p* < 0.10) the mRNA level of GBP1 in the ileum (Figure 9B) and significantly decreased (*p* < 0.05) the mRNA levels of IL-4 in the jejunum (Figure 9A) and mRNA levels of IL-β and IL-4 in the ileum (Figure 9B).

## 4. Discussion

PR, as the major bioactive ingredient derived from the root of the pueraria lobata, is commonly used for the treatment of fever, diarrhea, emesis, cardiac dysfunctions, liver injury, weight loss, and poisoning [40]. In an early study, PR was used as an antimicrobial agent [26]. So far, the effects of PR on piglets infected with ETEC K88 has not been reported. If the application of PR could effectively relieve the symptoms caused by ETEC infection in piglets, it would reduce the economic loss in the pig industry. This study investigated the protective effects of PR on ETEC K88-induced intestinal injury in piglets.

The villus is an important part of intestine, which plays an important role in the intestinal elimination of pathogens and the absorption of nutrients [41,42]. Additionally, the regression of the intestinal mucosa that is often manifested by shorter villi and deeper crypts is an obvious change in ETEC K88-infected piglets [17]. Villus height is usually a parameter used to determine intestinal health and intestinal absorptive functions [6], and the ratio of villus height to crypt depth is also an important indicator for assessing intestinal health and function [43]. Numerous studies have shown that decreased villus height and the ratio of villus height to crypt depth are associated with decreased growth performance and increased diarrhea rate in ETEC K88-challenged piglets [6]. In this study, ETEC K88 challenge significantly reduced the villus height and the surface area in the jejunum and ileum and the ratio of villus height to crypt depth in the ileum and jejunum and villus width in the jejunum; however, dietary PR supplementation effectively alleviated the decreased villus height, the ratio of villus height to crypt depth and the surface area of the jejunum and ileum in the piglets. The above results showed that the ETEC K88 challenge impaired intestinal morphology of jejunum and ileum, while PR supplementation alleviated the damage of jejunal and ileal intestinal morphology in ETEC K88-challenged piglets.

When piglets are born, the gastrointestinal tract is not colonized by bacteria. After birth, various microorganisms from the environment, breast milk and fecal–oral route will colonize in the intestine of piglets, and a stable microflora is slowly established in the gastrointestinal tract [2]. When piglets are in a healthy state, probiotics will build a microbial barrier by adhesion in the intestinal mucosa, which can resist the invasion of pathogenic microorganisms. It has been reported that microorganisms in the gastrointestinal tract can protect the intestinal barrier by resisting challenges from the environment and feed [44]. It is reported that when piglets are weaned, beneficial bacterium such as *Lactobacillus* and *Bifidobacterium* will be rapidly decreased, harmful bacteria, such as *Enterobacteriaceae* will be rapidly increased, and once the intestinal flora structure is disrupted, the structure and function of intestine are easily damaged in piglets [45]. *Lactobacillus* can resist pathogenic bacteria by secreting metabolites and degrading the macromolecular proteins into absorbable small-molecule peptides and amino acids in the feed in the intestinal tract [46]. It can also synthesize organic acids to accelerate intestinal peristalsis and promote the absorption of trace elements [46]. It was reported that the number of *Enterobacteriaceae* was significantly increased and the numbers of *Lactobacillus* and *Bifidobacterium* were significantly decreased in ETEC K88-challenged piglets [46]. In this experiment the number of *Enterobacteriaceae* was significantly increased in the jejunum and cecum, but the number of *Enterobacteriaceae* was significantly decreased in the colon in ETEC K88-challenged piglets. The number of *Lactobacillus* was significantly decreased in the jejunum, cecum and colon, while the number of *Bifidobacterium* in the colon was significantly decreased in ETEC K88-challenged piglets. PR supplementation significantly decreased the number of *Lactobacillus* in the jejunum, caecum and colon and significantly increased the number of *Bifidobacterium* in the colon in the ETEC + PR group. The above results suggested that PR supplementation could protect the piglet intestine by alleviating the decrease in beneficial bacterium.

Blood is an opaque red liquid circulating in the body. A change in blood composition can reflect the health state, metabolic levels, tissue and cell permeability. The premature weaning of piglets can cause changes in blood indicators. Alanine aminotransferase (ALT) and aspartate aminotransferase (AST) are intracellular enzymes of hepatocytes. When hepatocytes are damaged, ALT and AST will enter the blood; therefore, the activities of ALT and AST in blood can directly reflect the degree of hepatocyte damage [47]. Creatine kinase (CK) is an energy metabolism enzyme that ensures the energy supply of cellular tissues by promoting the generation of ATP by phosphocreatine [48]. A large number of studies have found that blood urea nitrogen (BUN) is negatively correlated with the growth rate and protein deposition of animal tissues, so the level of BUN can reflect the growth of tissue, protein deposition and feed utilization [49,50,51]. This study found that the plasma AST activity was significantly decreased after the ETEC K88 challenge, which is inconsistent with what Gong et al. reported [52]. This may be related to the duration of oral perfusion ETEC or individual animal differences. PR supplementation significantly increased the plasma CK activity, indicating that PR supplementation alleviated the blocked body energy metabolism caused by the ETEC K88 challenge.

BASO originates from precursor cells in the bone marrow, which enter the circulation and play a part in the function. Studies have found that the life cycle of mature BASO in health conditions is around 60 h, but when the organism is infected by pathogenic microorganisms and inflammation, its life cycle is prolonged [53]. After ETEC K88 infection in this study, the BASOR and the number of blood BASO were significantly increased in the ETEC group, and after PR supplementation, there was a significantly decreased the number of BASOR, which suggested that the inflammation that occurred in ETEC K88-challenged piglets and dietary PR supplementation relieved the inflammation of piglets challenged with ETEC K88.

After the ETEC K88 challenge, the immune organs or tissues of the body would produce a variety of cytokines, such as interleukin (IL) and tumor necrosis factor (TNF). IL-6 and TNF-α are well recognized as proinflammatory cytokine. Their main physiological role is to regulate the interactive response of various related cells in inflammation and affect the repair process after injury [54]. IL-6 is a phosphorylated and glycosylated cytokine that plays a huge role in the inflammatory response [53]. IL-6 plays an important role in the transition from acute inflammation to specific immunity or chronic inflammation [55]. The overproduction of TNF-α has been associated with multiple pathological processes, such as autoimmune diseases [56]. TNF-α can inhibit the growth of certain tumors, which acts as an immune modulator and a mediator of the inflammatory response and also plays a key role in the body’s resistance to infection [57]. It has been reported that the piglets develop inflammatory responses after being stimulated by LPS, and the contents of IL-6 and TNF-α in plasma are significantly increased [58]. Moreover, it has been reported that the mRNA levels of IL-1β, IL-6, IL-8, and TNF-α are upregulated in IPEC-J2 cells infected with PEDV [59]. Additionally, it was reported that the mRNA levels of the related inflammatory cytokines were increased in the PEDV-challenged Vero cells [60]. In this study, ETEC infection increased the plasma IL-6 and TNF-α contents in the piglets, and the administration of PR decreased the IL-6 and TNF-α in the plasma. These results suggested that ETEC infection could cause inflammation in piglets, but PR intervention could alleviate inflammation.

Oxidative stress is one of the main factors that impairs the integrity of the gastrointestinal barrier and increases intestinal permeability [61]. Hydrogen peroxide (H_2_O_2_) is ubiquitous in animals and is a highly oxidized product produced by cellular metabolism, which will cause damage when accumulated in the body [62]. Antioxidant enzymes are important components of the antioxidant system, and the antioxidant level of the host can be measured by measuring the activities of the antioxidant enzymes [63]. The antioxidant system is mainly composed of a variety of antioxidant enzymes, mainly including catalase (CAT), total superoxide dismutase (TSOD), and glutathione peroxidase (GSHPx) [64]. CAT is a common antioxidant enzyme in animals, which can catalytically decompose H_2_O_2_ into non-toxic H_2_O and O_2_ [65]. T-SOD can protect the cell membrane structure from lipid oxidation by removing the oxygen free radicals and other metabolic wastes generated by cell metabolism. The activity of T-SOD can better reflect the antioxidant capacity of the body [66]. Myeloperoxidase (MPO) is a heme protein and mainly exists in azurophil granules of neutrophils and monocytes, which is released upon cell activation into the phagolysosome or into the extracellular space [67]. In physiological cases, MPO consumes H_2_O_2_ and catalyzes the oxidation of chloride ions to generate hypochlorous acid (HClO) and the formation of oxidative free radicals, which has strong effects on killing of pathogenic microorganisms, cytotoxicity and detoxification [67]; however, HClO is a strong long-lived oxidant that can initiate lipid peroxidation reactions. When the body is in a state of inflammation and oxidative stress, the oxides catalyzed by MPO exceed the antioxidant capacity of the body, which will cause a variety of pathological processes and tissue damage, such as ischemic brain injury [68]. In this study, ETEC K88 challenge significantly decreased the activity of CAT and increased the content of MPO in the jejunum and ileum. The supplementation of PR significantly reduced H_2_O_2_ content in the jejunum and the activity of MPO in the jejunum and ileum and significantly increased the activities of CAT and T-SOD in the jejunum and ileum. The above results showed that the antioxidant capacity of jejunum and ileum in the piglets was reduced after the ETEC K88 challenge, and the supplementation of PR alleviated the reduction of intestinal antioxidant capacity after ETEC K88 challenge.

Interleukin-1 (IL-1) is mainly produced and secreted by epithelial cells and macrophages, with two existing forms of IL-1α and IL-1β. IL-1β is an important cytokine for regulating the inflammatory response in the animal body, which plays an important role in excluding the invasive pathogens and maintaining the balance of the body environment. IL-1β has an important role in immune processes, such as inflammatory response, immune response and tissue damage [69]. Interleukin-4 (IL-4) is one of the T helper (Th) 2 cytokines, which can multifunctionally regulate B cells and other non-immune cells [70]. It is an important regulator of the humoral immune response, which can reflect the cellular immunity and humoral immunity in animals [71]. MX protein is one of the antiviral proteins produced by host cells induced by type I interferon (IFN α/β). It has a wide range of antiviral effects. MX protein has inhibitory effects on a variety of negative-stranded RNA viruses, so IFN can directly resist negative-strand RNA viruses [72]. Myxovirus resistance protein 1 (MX1) has antiviral activity, and is closely related to the infection of the virus, which is very sensitive to the virus response or even a very small amount of the virus can induce cells to express MX1 protein. Recently, Myxovirus resistance protein 2 (MX2) has been reported to be a pan-herpesvirus restriction factor that interferes with early steps of herpesvirus replication [73]. Intercellular adhesion molecules (ICAM-1) and vascular cell adhesion molecules (VCAM-1) play an important role in the adhesion of inflammatory cells to the vascular endothelium and in the transmembrane transport of inflammatory cells [74]. Guanylate-binding protein-1 (GBP1) and guanylate-binding protein-2 (GBP2) belong to the member of guanylate-binding proteins superfamily for immunity against microorganisms and viral pathogens. GBP1 is a guanosine-5′-triphosphate-binding protein in the dynamin superfamily and regulates multiple cell functions [75,76]. Furthermore, high expression of GBP1 and GBP2 predicted poor prognosis in survival analysis of clinical samples [75,76]. In this study, the relative expression level inflammatory-related genes (IL-1 β and IL-4) was upregulated after the ETEC K88 challenge in the jejunum and ileum, the relative expression levels of genes for immune-related and antiviral genes (ICAM1, VCAM1, MX1, MX2, GBP1, GBP2) in the jejunum and the relative expression levels of genes for immune-related and antiviral genes (ICAM1, MX1, MX2) in the ileum were downregulated. After PR intervention, the relative gene expression levels of piglets with jejunal IL-4 and ileal IL-1β and IL-4 was downregulated; gene expression levels of ICAM1, VCAM1, MX1, MX2 and GBP1 were upregulated in the jejunum of piglets; and gene expression levels of MX2 and GBP1 in the piglet ileum, while the relative expression levels of MX2 and GBP1 were upregulated in the ileum. The above results indicated that the ETEC K88 challenge stimulated the production of inflammatory cytokines in the small intestine of piglets while inhibiting the immune function of the small intestine and eventually led to intestinal injury. The administration of PR partially inhibited the production of inflammatory cytokines in the small intestine and improved the immune function of the jejunum and ileum, thus alleviating intestinal damage.

## 5. Conclusions

In conclusion, dietary supplementation with 5 mg/kg PR alleviated intestinal injury and improved the intestinal function of piglets challenged with ETEC K88. PR may improve the intestinal function of piglets challenged with ETEC K88 by enhancing the anti-inflammatory functions (indicated by improving the plasma biochemical index, immune cell numbers and cytokines levels), improving its antioxidative capacity (indicated by improving redox status and attenuating oxidative damage), and enhancing the intestinal mucosal barrier. These results provide important insights into the development of an effective treatment for ETEC K88 infection and other enteric diseases.

## Figures and Tables

**Figure 1 animals-13-01908-f001:**
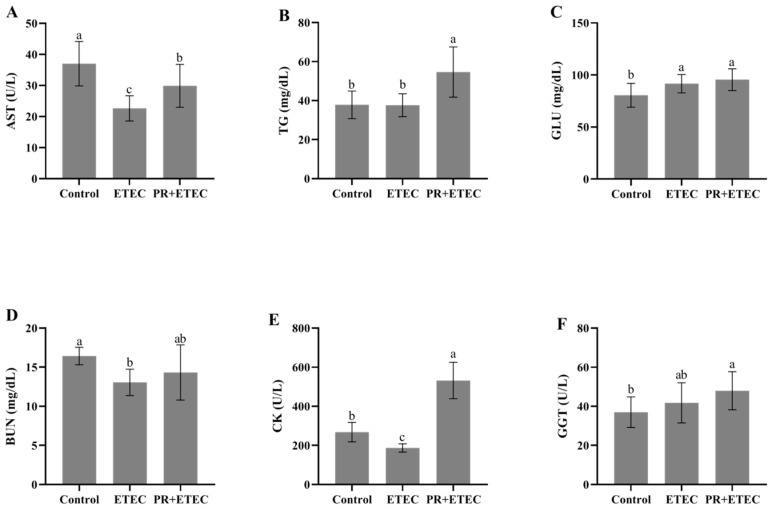
Effects of PR supplementation on the plasma biochemical indices in piglets challenged with ETEC. Values are represented as the mean ± SD, n = 8. ^a, b, c^: Values within a column not sharing a common superscript letter indicate a significant difference at *p* < 0.05. (**A**) Activities of plasma glutamate (AST); (**B**) triglyceride (TG); (**C**) glucose (GLU); (**D**) blood urine nitrogen (BUN); (**E**) creatine kinase (CK); (**F**) glutamyl transpeptidase (GGT).

**Figure 2 animals-13-01908-f002:**
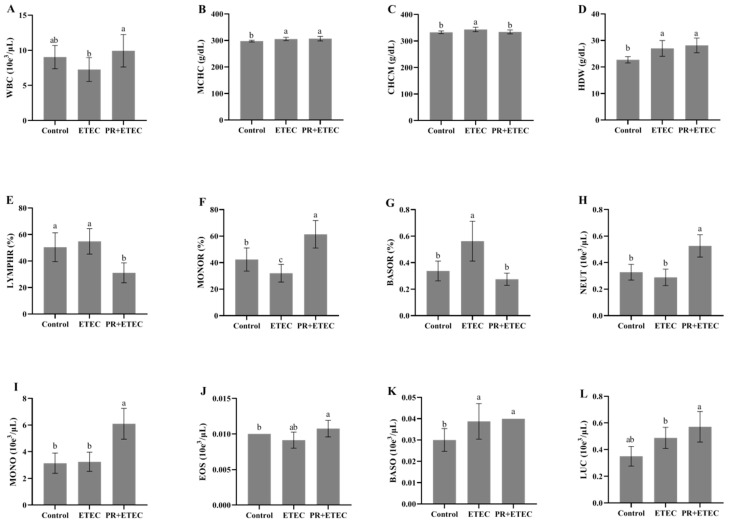
Effects of PR supplementation on the blood cell counts of piglets after ETEC K88 infection. Values are represented as the mean ± SD, n = 8. ^a, b, c^: Values within a column not sharing a common superscript letter indicate a significant difference at *p* < 0.05. (**A**) White blood cells (WBCs); (**B**) mean corpuscular hemoglobin concentration (MCHC); (**C**) corpuscular hemoglobin concentration mensuration (CHCM); (**D**) hemoglobin distribution width (HDW); (**E**) percentage of lymphocyte (LYMPHR); (**F**) percentage of monocyte (MONOR); (**G**) percentage of blood basophils (BASOR); (**H**) neutrophil (NEUT); (**I**) monocyte (MONO); (**J**) eosinophilic granulocyte (EOS); (**K**) basophils (BASO); (**L**) large unstained cell (LUC).

**Figure 3 animals-13-01908-f003:**
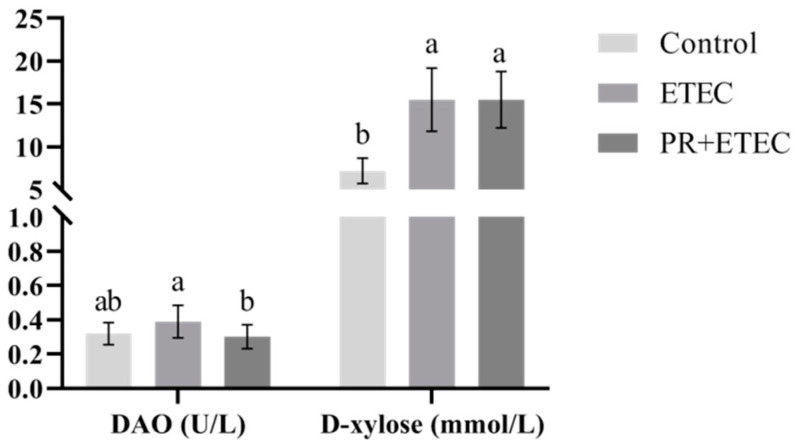
Effects of PR supplementation on DAO activity and D-xylose concentration in piglets after ETEC K88 infection. Values are represented as the mean ± SD, n = 8. ^a, b^ Values within a column not sharing a common superscript letter indicate a significant difference at *p* < 0.05.

**Figure 4 animals-13-01908-f004:**
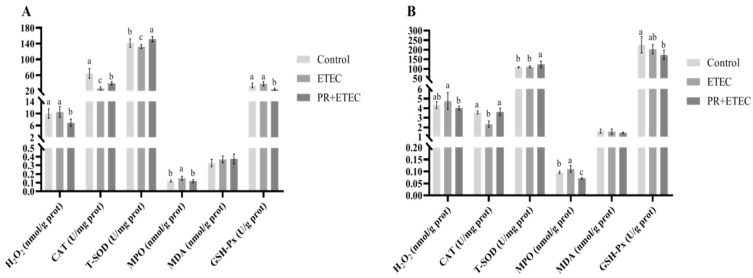
Effects of PR supplementation on the redox status in piglets after ETEC K88 infection in the jejunum and ileum. Values are represented as the mean ± SD, n = 8. ^a, b, c^: Values within a column not sharing a common superscript letter indicate a significant difference at *p* < 0.05. (**A**) Jejunum; (**B**) ileum.

**Figure 5 animals-13-01908-f005:**
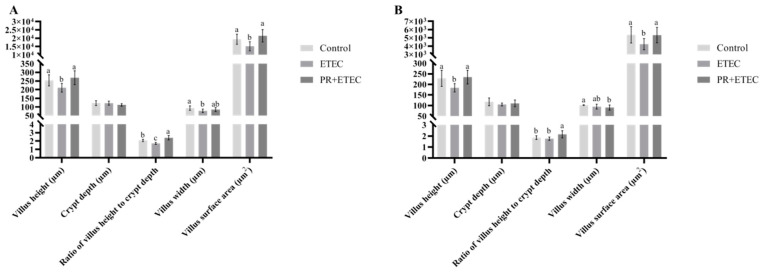
Effects of PR supplementation on the intestinal mucosal morphology of pigs after ETEC K88 infection. Values are represented as the mean ± SD, n = 8. ^a, b, c^: Values within a column not sharing a common superscript letter indicate a significant difference at *p* < 0.05. (**A**) Jejunum; (**B**) ileum.

**Figure 6 animals-13-01908-f006:**
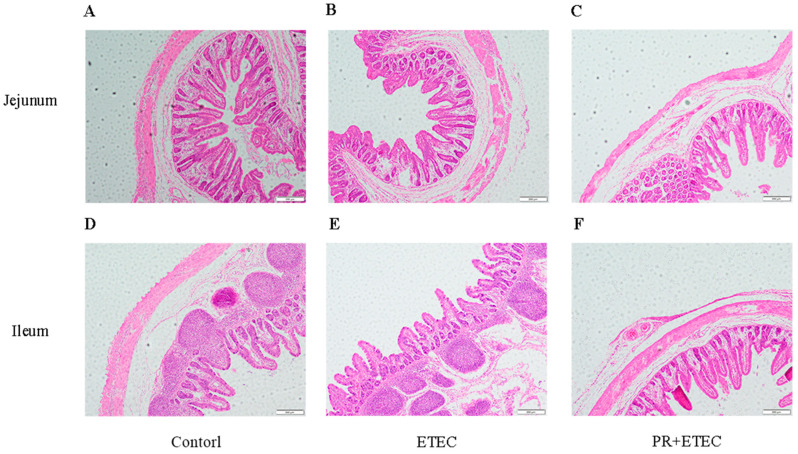
Histopathological analysis of jejunum and ileum from different groups. (**A**) Jejunum in the control group; (**B**) jejunum in the ETEC group; (**C**) jejunum in the PR + ETEC group; (**D**) ileum in the control group; (**E**) ileum in the ETEC group; (**F**) ileum in the PR + ETEC group.

**Figure 7 animals-13-01908-f007:**
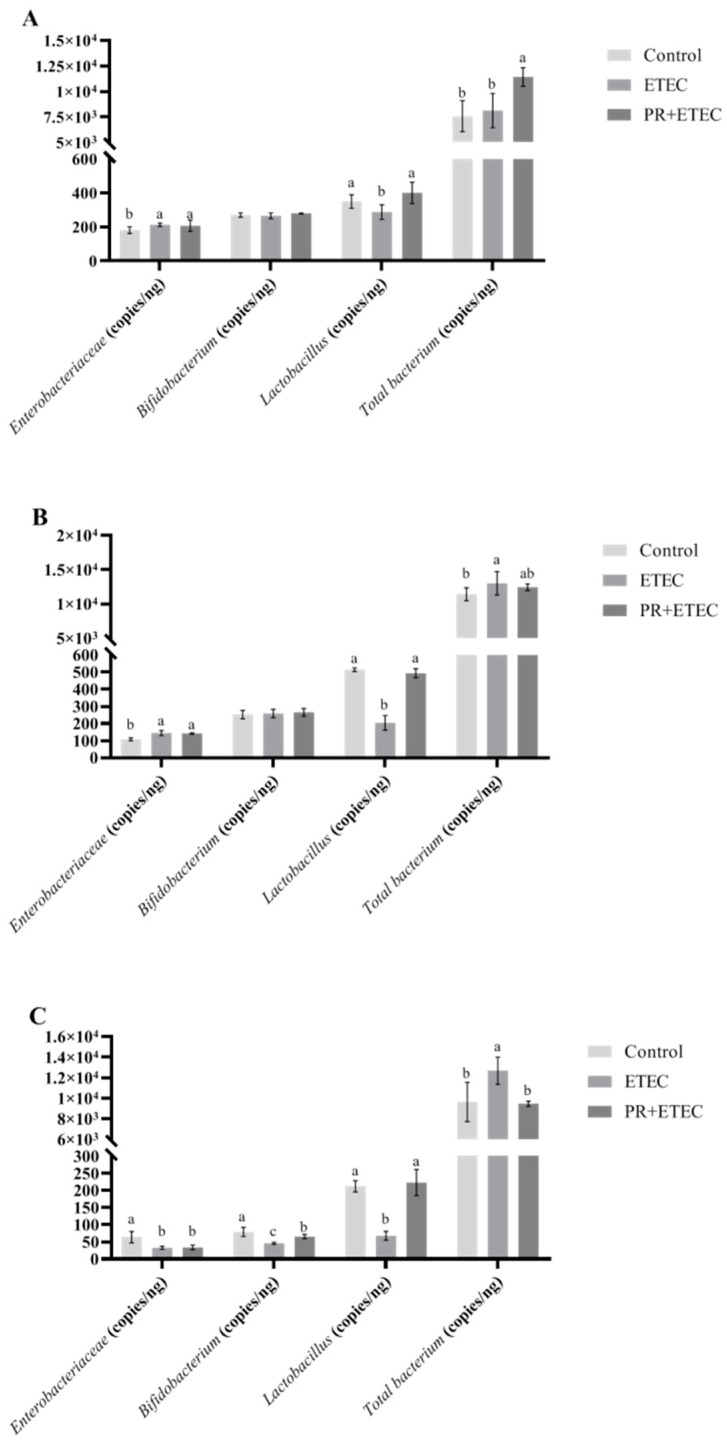
Effects of PR supplementation on the intestinal microflora of piglets. Values are represented as the mean ± SD, n = 8. ^a, b, c^ Values within a column not sharing a common superscript letter indicate a significant difference at *p* < 0.05. (**A**) Jejunum; (**B**) caecum; (**C**) colon.

**Figure 8 animals-13-01908-f008:**
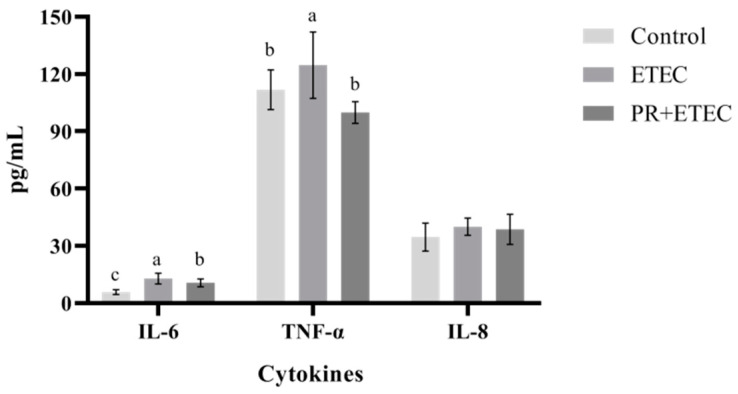
Effects of PR supplementation on cytokines in the plasma of piglets after ETEC infection. Values are represented as the mean ± SD, n = 8. ^a, b, c^: Values within a column not sharing a common superscript letter indicate a significant difference at *p* < 0.05.

**Figure 9 animals-13-01908-f009:**
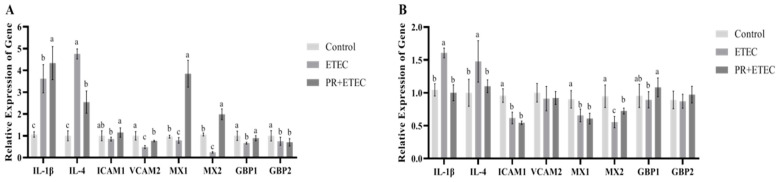
Effects of PR supplementation on mRNA levels of genes associated in the jejunum and ileum of piglets after ETEC K88 infection. Values mean ± SD, n = 8. ^a, b, c^: Values within a column not sharing a common superscript letter indicate a significant difference at *p* < 0.05. (**A**) Jejunum; (**B**) ileum.

**Table 1 animals-13-01908-t001:** Composition and nutrient contents of the diet (air-dried basis) %.

Items	Content (%)
Carbohydrate	≥46
Crude protein	≥38
Lactobiose	≥20
Water	≤9.0
Ash	≤8.0
Lysine	≥2.75
Crude fat	≥2.0
Threonine	≥1.7
Methionine	≥0.7
Ca	0.3–0.8
Total phosphorus	0.4–0.9

Contents of vitamins and minerals per kilogram of diet: Fe 100 mg; Cu 150 mg; Mn 40 mg; Zn 100 mg; I 0.5 mg; Se 0.3 mg; VA 1800 IU; VD3 4000 IU; VE 40 IU; VK3 4 mg; VB1 6 mg; VB2 12 mg; VB6 6 mg; VB12 0.05 mg; biotin 0.2 mg; folic acid 2 mg; nicotinic acid 50 mg; and calcium pantothenate 5 mg.

**Table 2 animals-13-01908-t002:** Sequences of the primers used for digital PCR analysis.

Gene	Forward (5′-3′)	Reverse (5′-3′)
*Enterobacteriaceae*	CATGCCGCGTGTATGAAGAA	CGGGTAACGACAATGAGCAAA
*Bifidobacterium*	TCGCGTC(C/T)GGTGTGAAAG	CCACATCCAGC(A/G)TCCAC
*Lactobacillus*	AGCAGTAGGGAATCTTCCA	CACCGCTACACATGGAG
*Clostridium*	AATGACGGTACCTGACTAA	CTTTGAGTTTCATTCTTGCGAA
Total bacterium	CGGTCCAGACTCCTACGGG	TTACCGCGGCTGCTGGCAC

**Table 3 animals-13-01908-t003:** Sequences of the primers used for quantitative RT-PCR analysis.

Gene	Forward (5′-3′)	Reverse (5′-3′)
*RPL4*	GAGAAACCGTCGCCGAAT	GCCCACCAGGAGCAAGTT
*MX1*	AGTGCGGCTGTTTACCAAG	TTCACAAACCCTGGCAACTC
*MX2*	CGCATTCTTTCACTCGCATC	CCTCAACCCACCAACTCACA
*IL-1β*	CAACGTGCAGTCTATGGAGT	GAGGTGCTGATGTACCAGTTG
*IL-4*	AGGAGCCACACGTGCTTGA	TTGCCAAGCTGTTGAGATTCC
*ICAM1*	ACCCACCCACACCTTGCTAC	TCACATTCTTCTTTGTCACCACCT
*VCAM1*	GGATGGTGTTTGCCGTTTCT	CTGGTCCCGTTAGTTTTCACTTTT
*GBP1*	TGGACTTGGAAACAGATGGAGA	GGATACAGAGTCGAGGCAGGTT
*GBP2*	ACCAGGAGGTTTTCGTCTCTCTATT	TCCTCTGCCTGTATCCCCTTT

**Table 4 animals-13-01908-t004:** Effects of PR supplementation on the growth performance of piglets after ETEC K88 infection.

Item	Control	ETEC	PR + ETEC
Average daily feed intake (g/d)			
Day 4–7	498.5 ± 66.7	495.1 ± 67.0	496.8 ± 50.1
Day 8–9	571.0 ± 34.5 ^b^	398.7 ± 88.8 ^a^	458.6 ± 66.5 ^ab^
Average daily gain (g/d)			
Day 4–10	26.96 ± 20.47 ^a^	−10.18 ± 18.93 ^b^	−6.61 ± 38.39 ^b^

Values are represented as the mean ± SD, n = 8. ^a, b^ Values within a column not sharing a common superscript letter indicate a significant difference at *p* < 0.05.

**Table 5 animals-13-01908-t005:** Effect of PR on fecal scores of ETEC-infected piglets.

Groups	Fecal Scores				Weighted Mean
	0	1	2	3	
Control	42	8	6	0	0.36 ^a^
ETEC	24	15	24	7	1.2 ^b^
PR + ETEC	29	5	16	6	0.98 ^b^

Values are represented as the weighted mean, n = 8. ^a, b^ Values within a column not sharing a common superscript letter indicate a significant difference at *p* < 0.05.

## Data Availability

The datasets used and/or analyzed during the current study are available from the corresponding authors on reasonable request.

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
