# Peer review of "Dietary Supplementation with Puerarin Improves Intestinal Function in Piglets Challenged with Escherichia coli K88"

_animals, 2023, doi:10.3390/ani13121908_

Round 1

Reviewer 1 Report

This report showed that dietary supplementation with puerarin alleviated ETEC K88-induced intestinal injury by improve intestinal function in piglets. The study focus on an important topic and the organization of the manuscript is appropriate, but several issues need to address.

1.         We know that weaning stress is a big issue for intestinal injury, why choose suckling piglets weaned at 7 days instead of weaning piglets at normal weaning days such as 21-28 days to evaluate the effect of dietary supplementation with PR in this study? this need to highlight in the introduction.

2.         Has the diarrhea rate of piglets been measured in the study? An important indicator for the success of ETEC K88 challenged model is the increase of the diarrhea rate.

3.         In Table 4, it is better to use dry matter to show the ADFI because there is a lot water in the diet, and to show the ADG for the piglets. The ADG is about only 20 g according to their body weight for piglets, this is so low compare with suckling piglets, are the piglets normal?

4.         For data analysis, what’s method used for multiple comparison?

5.         Revise English writing and wording, such as Line 15, 76, 85, 242, etc..

Author Response

Response to Reviewer 1:

Thank you very much for your valuable comments on our manuscript. We have carefully examined and revised the manuscript according to your comments in the revised manuscript.

Reviewer 1

  1. We know that weaning stress is a big issue for intestinal injury, why choose suckling piglets weaned at 7 days instead of weaning piglets at normal weaning days such as 21-28 days to evaluate the effect of dietary supplementation with PR in this study? this need to highlight in the introduction.

Response:Thanks for the reviewer’s valuable advice. This study focuses on the effect of PR on intestinal function in piglets infected with ETEC K88. In this study, the intestinal injury model of ETEC K88 infected piglets is successfully established in our previous study, and suckling piglets weaned at 7 days were selected as the research object in this model (Wu et al. 2021). This study is based on the intestinal injury model of ETEC K88 infected piglets to evaluate the effect of dietary supplementation with PR, so suckling piglets weaned at 7 days are selected in this study. And relevant informations have been added to the text (Lines 53 to 55). 

Wu T, Shi Y, Zhang Y, Zhang M, Zhang L, Ma Z, Zhao D, Wang L, Yu H, Hou Y, Gong J. Lactobacillus rhamnosus LB1 Alleviates Enterotoxigenic Escherichia coli-Induced Adverse Effects in Piglets by Improving Host Immune Response and Anti-Oxidation Stress and Restoring Intestinal Integrity. Frontiers In Cellular And Infection Microbiology. 2021, 11, 724401.

  1. Has the diarrhea rate of piglets been measured in the study? An important indicator for the success of ETEC K88 challenged model is the increase of the diarrhea rate.

Response:Thanks for your careful work. We have been measured the diarrhea rate of piglets in the study, and the diarrhea rate of piglets have been added to the revised manuscript (Line 204 to 215).

  1. In Table 4, it is better to use dry matter to show the ADFI because there is a lot water in the diet, and to show the ADG for the piglets. The ADG is about only 20 g according to their body weight for piglets, this is so low compare with suckling piglets, are the piglets normal?

Response:Thanks for the reviewer’s valuable advice. In Table 4, it is used dry matter to show the ADFI, because each group has eight piglets, a piglet is kept five meals a day, one meal contains 20 g of dry matter. Each day, 800 g of dry matter will be provided to each group in the diet, so it is used dry matter to show the ADFI. During the 3-day adaptation period, it is not within the formal trial, so ADFI was not recorded until the formal trial began on the fourth day, the period of formal trial for this trial is 7 days (D4-D10). After the modifications, the ADG is about 26 g, and update the data in Table 4. I agree with your comment about the lower ADG. Based on the previous published literatures about healthy 7-day-old piglets. After the adaptation period, the mean weight of piglets is 3.17 ± 0.51 kg, ADG is about 80 g (Wu, et al, 2021); the mean weight of piglets is 3.17 ± 0.25 kg, ADG is about 50 g (Wu, et al, 2020). The piglets are healthy during the trial, but the piglets have a light initial weight (1.79 ± 0.24 kg). The poor body weight may explain the larger difference in ADG compared to previously published papers (Wu, et al, 2021; Wu, et al, 2020; Zhang, et al, 2020; Wu, et al, 2023). The weight of the seventh day of weaning piglets should be provided in the previous amendment, and the data has been placed in the material method (Line114-115).

Wu M, Yi D, Zhang Q, et al. Puerarin enhances intestinal function in piglets infected with porcine epidemic diarrhea virus. Sci Rep. 2021;11(1):6552. Published 2021 Mar 22. doi:10.1038/s41598-021-85880-5

Wu M, Zhang Q, Yi D, et al. Quantitative Proteomic Analysis Reveals Antiviral and Anti-inflammatory Effects of Puerarin in Piglets Infected With Porcine Epidemic Diarrhea Virus. Front Immunol. 2020;11:169. Published 2020 Feb 26. doi:10.3389/fimmu.2020.00169

Zhang Z, Wang S, Zheng L, et al. Tannic acid-chelated zinc supplementation alleviates intestinal injury in piglets challenged by porcine epidemic diarrhea virus. Front Vet Sci. 2022;9:1033022. Published 2022 Oct 10. doi:10.3389/fvets.2022.1033022

Wu T, Zhang Q, Xu H, et al. Protective effects of α-terpineol and Bacillus coagulans on intestinal function in weaned piglets infected with a recombinant Escherichia coli expressing heat-stable enterotoxin STa. Front Vet Sci. 2023;10:1118957. Published 2023 Feb 10. doi:10.3389/fvets.2023.1118957

  1. For data analysis, what’s method used for multiple comparison?

Response:Thanks for the reviewer’s valuable advice. The data is analyzed by using the one-way ANOVA procedure in the SPSS 25.0 software (SPSS Inc. Chicago, USA) and expressed as mean ± SD. The differences between means among the treatment groups were determined by the Duncan's multiple range test.  

  1. Revise English writing and wording, such as Line 15, 76, 85, 242, etc..

Response:Thanks for the reviewer’s valuable advice. Thanks for your careful work. Modification has been made.

Line 16: “...killed0...” to “...killed...”

Line 78: “PR was used as an antimicrobial agent...” to “PR was used to an antimicrobial agent...”

Line 91: “...the The...” to “...the...”

Line 260: “Data on activities of T-SOD...” to “The activities of T-SOD”

Reviewer 2 Report

The aim of the article ‘’ Dietary supplementation with puerarin improves intestinal function in piglets challenged with Escherichia Coli K88’’ was to investigate the effect of puerarin supplementation on 9 growth performance and intestinal function in piglets challenged with enterotoxigenic Escherichia 10 coli (ETEC) K88. This study deals with an interesting topic and fits with the journal's scope. This is a well-designed and well-written work, providing interesting results and novel results.

Author Response

Response to Reviewer 2:

Thank you very much for your valuable comments on our manuscript. We have carefully examined and revised the manuscript according to your comments in the revised manuscript.

Reviewer 2

  1. In the keywords place piglets in the first place, it is the focus species of the study.

Response:Thanks for the reviewer’s valuable advice. In the keywords place have been placed piglets in the first place.

  1. provide details about the litter characteristics to which the trial piglets belonged

(number of total born piglets, dead piglets, stillbirth etc)

Response:Thanks for your careful work. Piglets are purchased in a large commercial sow farm, many sows deliver on the same day. The number of piglets born on the same day far exceeded the trial requirement, and the healthy piglets from the trial are randomly selected from them. So, we do not have much knowledge about the litter characteristics of piglets (number of total born piglets, dead piglets, stillbirth etc).

  1. provide details about the vaccination program of the sows from which the trial piglets were selected

Response:Thanks for the reviewer’s valuable advice. The vaccination program of the sows were about pseudorabies virus vaccine, swine fever vaccine, porcine circovirus type 2 (PCV-2) vaccine, porcine circovirus vaccine.

  1. the farrowing date (114th day) was the same for all sows or you induced the farrowing ?

Response:Thanks for the reviewer’s valuable advice. The piglets are purchased from a large farm and they are artificially bred by the farm, the number of piglets meet the study requirements, and the farrowing date (114th day) is the same for all sows. Many sows are delivered on the same day, and we randomly buy healthy piglets that are delivered on the same day. The experimental piglets are delivered on the same day, so the sows did not induce the farrowing.

  1. provide more data on the euthanasia of trial piglets

Response:Thanks for the reviewer’s valuable advice. The data have been added to the trial design (Line 126-128).

  1. The piglets in each pen had possible physical contact with piglets from another pen, both within treatment or from different treatments?

Response:Thanks for the reviewer’s valuable advice. All pigs were housed individually in stainless steel metabolic cages (1.0 × 1.5 m2), maintained at an ambient temperature of 22-25 °C in an environmentally controlled room. Using metabolic cages can avoid the contact of piglets with each other in different treatment conditions.

  1. on which criteria was the selection of the number of piglets that would participate in the study based.

Response:Thanks for your careful work. The number of piglets refered to the other experimental designs of our research group, and the relevant references are as follows.

[1]  Zhang Y, Yi D, Xu H, Tan Z, Meng Y, Wu T, Wang L, Zhao D, Hou Y. Dietary supplementation with sodium gluconate improves the growth performance and intestinal function in weaned pigs challenged with a recombinant Escherichia coli strain. BMC Vet Res. 2022 Aug 6;18(1):303. doi: 10.1186/s12917-022-03410-5. PMID: 35933350; PMCID: PMC9356463.

[2]  Wu T, Shi Y, Zhang Y, Zhang M, Zhang L, Ma Z, Zhao D, Wang L, Yu H, Hou Y, Gong J. Lactobacillus rhamnosus LB1 Alleviates Enterotoxigenic Escherichia coli-Induced Adverse Effects in Piglets by Improving Host Immune Response and Anti-Oxidation Stress and Restoring Intestinal Integrity. Front Cell Infect Microbiol. 2021 Nov 2;11:724401. doi: 10.3389/fcimb.2021.724401. PMID: 34796123; PMCID: PMC8594739.

[3]  Zhang Y, Yi D, Xu H, Tan Z, Meng Y, Wu T, Wang L, Zhao D, Hou Y. Dietary supplementation with sodium gluconate improves the growth performance and intestinal function in weaned pigs challenged with a recombinant Escherichia coli strain. BMC Vet Res. 2022 Aug 6;18(1):303. doi: 10.1186/s12917-022-03410-5. PMID: 35933350; PMCID: PMC9356463.

[4]  Zhang Z, Wang S, Zheng L, Hou Y, Guo S, Wang L, Zhu L, Deng C, Wu T, Yi D, Ding B. Tannic acid-chelated zinc supplementation alleviates intestinal injury in piglets challenged by porcine epidemic diarrhea virus. Front Vet Sci. 2022 Oct 10;9:1033022. doi: 10.3389/fvets.2022.1033022. PMID: 36299630; PMCID: PMC9589514.

[5]  Li H, Zhang Y, Xie J, Wang C, Yi D, Wu T, Wang L, Zhao D, Hou Y. Dietary Supplementation with Mono-Lactate Glyceride Enhances Intestinal Function of Weaned Piglets. Animals (Basel). 2023 Apr 11;13(8):1303. doi: 10.3390/ani13081303. PMID: 37106866; PMCID: PMC10135088.

  1. Table 4: write in the title the full concept and not only the abbreviation (like in ADFI).

Response:Thanks for the reviewer’s valuable advice. Modifications have been made in Table 4.

  1. you could discuss the economic impact of your results on pig production.

Response:Thanks for the reviewer’s valuable advice. Effects on pig production have been added (Line 343 to 344).

  1. Minor comments

Response:Thanks for the reviewer’s valuable advice. The questions raised have been modified accordingly.

L28: “bacterium” to “.. bacteria..”

L71: “..the intestinal” to “.. intestinal”

L84: “.. improve” to “.. improved..”

L97:  “..eonate” to “neonate” 

L148: “.. paraffifin..” to “paraffin..”

L149: “..crypt..” to “ the crypt..”

L203: “..significant” to “a significant...”

L218: “.. glutamase..” to “.. glutamate..”

L222: “..of and triglyceride..” to “..of triglyceride..”

L226: “.. Efects..” to “..Effects”

L228: “.. significant diference..” to “.. significant difference..”

L233: “... concerntration..” to “. concentration..” 

L234: “... concerntration..” to “. concentration..”

L243: “... concerntration..” to “. concentration..”

L244: “... concerntration..” to “. concentration..”

L257: “significant..” to “ a significant..”

L261: “.. control group..” to “ .. the control group..”

L266: “.. jejunum..” to “ .. the jejunum.”

L272:“significant..” to “ a significant..”

L309: “.. intestinal..” to “..the intestinal..”

L315: “.. was significantly increased..” to “..were significantly increased..”

L323: “..were show..” to “....were shown..”

L336:“significant..” to “ a significant..”  

L341: “..early study..” to “..an early study..”  

L351: “..function..” to “..functions..”  

L356: “..surface area...” to “..the surface area...”

L368: “..intestinal mucosa..” to “..the intestinal mucosa..

L370: “..challenge..” to “..challenges..”

L389: “..piglet intestine..” to “..the piglet intestine..”

L407: “.can resist to the invasion of the pathogens..” to “..can resist the invasion of pathogens..”

L392: .. “to affect”..to “affect”

L426: “..is able to inhibit..” to “..can inhibit..”

L443: “. host..” to “....the host..”

Round 2

Reviewer 1 Report

The authors answered the questions raised by this reviewer, however, it needs to show the information in the manuscript, such as the method for multiple comparisons. It also needs to provide the information for the fecal scores (how to define the scores) in the Material and Methods and the statistical method to analyze the fecal scores.

Author Response

Reviewer 1

  1. it needs to show the information in the manuscript, such as the method for multiple comparisons.

Response:Thanks for the reviewer’s valuable advice. The method for multiple comparisons have been added to the Statistical Analysis section (Line 184-188).

  1. It also needs to provide the information for the fecal scores (how to define the scores) in the Material and Methods and the statistical method to analyze the fecal scores.

Response:Thanks for your careful work. The information for the fecal scores have been provided in Line 1074-106. The statistical method to analyze the fecal scores has been added to lines 186-187.